# Transcriptomic and Metabolomic Analyses Providing Insights into the Coloring Mechanism of *Docynia delavayi*

**DOI:** 10.3390/foods11182899

**Published:** 2022-09-18

**Authors:** Yuchang Wang, Yuyang Song, Dawei Wang

**Affiliations:** 1Key Laboratory for Forest Resources Conservation and Utilization in the Southwest Mountains of China Ministry of Education, Southwest Forestry University, Kunming 650224, China; 2Key Laboratory for Forest Genetic and Tree Improvement and Propagation in Universities of Yunnan Province, Southwest Forestry University, Kunming 650224, China; 3Department of Forestry, Agricultural College, Xinjiang Shihezi University, Shihezi 832003, China

**Keywords:** *Docynia delavayi*, fruit color, anthocyanin, metabolome, transcriptome

## Abstract

The metabolome and transcriptome profiles of three different variations of mature *Docynia delavayi* fruit were synthesized to reveal their fruit color formation mechanism. A total of 787 secondary metabolites containing 149 flavonoid metabolites, most of which were flavonoids and flavonols, were identified in the three variations using ultra performance liquid chromatography- tandem mass spectrometry (UPLC-MS/MS), and we found that the secondary metabolites cyanidin-3-O-galactoside and cyanidin-3-O-glucoside were the major coloring substances in *D. delavayi*. This was associated with the significant upregulation of the structural genes *F3H* and *F3′H* in the anthocyanin synthesis pathway and the control genes WRKY, MYB, bZIP, bHLH, and NAC in RP. *F3′H* expression may play a significant role in the selection of components for anthocyanin synthesis. Our results contribute to breeding and nutritional research in *D. delavayi* and provide insight into metabolite studies of the anthocyanin biosynthetic pathway.

## 1. Introduction

*Docynia delavayi* is an endemic plant of the Rosaceae family found on the Yunnan–Kweichow plateau in China. *D. delavayi* is highly adaptable, and its fruits are rich in nutrients, making it a famous local fruit. The destruction of the environment, the loss of wild resources, and the failure to reasonably apply germplasm resources has affected the development *D. delavayi* cultivation. Previous studies have shown that the antioxidants in the bark can scavenge free radicals from cigarettes, effectively reducing the harmful effects of smoking [1]. The fruit of *D. delavayi* is rich in flavonoids, alkaloids, polysaccharides, and other bioactive substances, and its extract has good antioxidant and anti-tumor properties [2], making it a wild fruit tree with great potential for exploitation. Identifying and elucidating the effective metabolites of *D. delavayi* will help to develop and exploit it.

The pericarp color leaves a visual impression on consumers and is an important indicator of fruit selection. The bright color of the pericarp attracts animals to spread its seeds, and the pericarp also helps the plant to resist abiotic stress [3,4]. In prolonged investigations and research, it was found that *D. delavayi* has rich phenotypic variation, which is especially evident in the color of its fruit peels. *D. delavayi* traditionally has a bright green pericarp (GP) or yellow pericarp (YP) (Figure 1). Over the course of our investigations, we found a red bud variant and obtained stable genetic traits (fruit ripening to red pericarp) through asexual reproduction (Figure 1). Understanding the genetic basis of the traits associated with the color of the skin of *D. delavayi* fruit will help us to breed it to be more attractive and healthier for consumers.

The fruit’s color can mainly be attributed to anthocyanins, chlorophylls, and carotenoids [5,6]. Anthocyanins are the main secondary metabolites that determine fruit color. During anthocyanin biosynthesis, a series of metabolite alterations take place, and a number of genes are regulated, after which the anthocyanins undergo glycosylation, methylation, and acylation to form stable anthocyanins, which are then transported into plant cells [7,8]. Previous studies have established that brightly colored fruits tend to exhibit high expression of key downstream genes, such as the structural genes *ANS*, *DFR*, and *UFGT,* in the anthocyanin biosynthetic pathway [9]. Several reports have shown that transcription factors (TFs) also play an important role in the anthocyanin pathway, regulating color changes and the biological activity of metabolites [10]. The synthetic pathway of anthocyanins is mainly regulated by the MYB, bHLH, and WD40 (MBW) complexes [11,12]. The pathway is a key regulator of anthocyanin accumulation activation in a variety of plants [12,13]. However, other TFs (bZIP, WRKY, HD-Zip I, HD-Zip II, MADS, EFR, and NAC) are also implicated in anthocyanin regulation [14,15].

Metabolomics provides a technique that can reflect nutritional levels and physiological status, which can serve as a bridge between metabolites and a certain phenotype [16]. Transcriptome sequencing can obtain transient, high-throughput gene expression of living organisms, which, in turn, reveals the intrinsic connection between genes and life phenomena [17,18]. Studies have been conducted on the anthocyanin biosynthesis pathways in other fruits [19,20], but a study on the coloring mechanism of *D. delavayi* has not yet been reported. In this study, metabolomic and transcriptomic approaches were used to analyze the metabolic components and molecular mechanisms of *D. delavayi*. The research reveals the metabolic pathways and candidate genes for color change in *D. delavayi* fruits and provides a new perspective on fruit coloration and the selection of new cultivars.

## 2. Materials and Methods

### 2.1. Sample Collection and Storage

In our study, we collected three types (RP, YP, and GP) of mature *D. delavayi* (Figure 1). Samples were selected from plants with normal growth and no obvious pests or diseases. We selected 30 fruits of equal maturity and with a similar color, shape, and size. The fruits were rinsed with sterile water and dried. Fruit peel samples (0.1 cm thick) were collected from several *D. delavayi* (Rosaceae) trees that were approximately 30 years old and growing in Lancang (22°29′55″ N, 100°14′59″ E, 1950 m; Yunnan, China) in October 2020. The fruit peels from each of the three fruit trees were evenly mixed as one biological replicate to create a total of three biological replicates. Three biological replicates were selected for each plant type. Fruit peels were cut off and immediately frozen in liquid nitrogen and stored at −80 °C until further analyses were carried out.

### 2.2. Quantification of Total Flavonoids and Anthocyanins

Total flavonoid sampling was carried out using a spectrophotometer method [21]. The flavonoids formed a red complex with a characteristic absorption peak at 470 nm with aluminum ions in alkaline nitrite solution. The total flavonoid content of the sample can be calculated by measuring the absorbance value of the sample extract at 470 nm. The anthocyanin content was assayed according to Li et al. [22]. The anthocyanin concentration was measured using an ultraviolet spectrophotometer at λ620 and λ650. The anthocyanin concentration was calculated using the Arnon equation: C = ((A530 − A620) − 0.1(A650 − A20))/ε × (V/m) × 10^6^ × M × 10^−3^, where C is the anthocyanin concentration (ng/g), A is the absorbance at the corresponding wavelength, ε is the anthocyanin extinction coefficient (4.62 × 10^4^), V is the volume of the extracting solution (mL), m is the weight of the fresh pericarp (g), and M is the molecular weight of anthocyanin (287.24).

### 2.3. Metabolite Identification and Data Analysis

#### 2.3.1. Sample Preparation and Extraction

The pericarps of three types (RP, YP, and GP) of mature *D. delavayi* were freeze-dried by vacuum freeze-dryer (Scientz-100F)(Verder Shanghai Instruments and Equipment Co., Ltd., Shanghai, China). The freeze-dried sample was crushed using a mixer mill (MM 400, Retsch) (Verder Shanghai Instruments and Equipment Co., Ltd., Shanghai, China) with a zirconia bead for 1.5 min at 30 Hz. Then, 100 mg of lyophilized powder was dissolved with 1.2 mL of a 70% methanol solution, vortexed for 30 s every 30 min for 6 times in total, and the samples placed in a refrigerator at 4 °C overnight. Following centrifugation at 12,000 rpm for 10 min, the extracts were filtrated (SCAA-104, 0.22 μm pore size; ANPEL, Shanghai, China, http://www.anpel.com.cn/, accessed on 4 February 2021) before UPLC-MS/MS analysis.

#### 2.3.2. UPLC Conditions

The sample extracts were analyzed using an UPLC-ESI-MS/MS system (UPLC, SHIMADZU Nexera X2, www.shimadzu.com.cn/, accessed on 10 February 2021; MS, Applied Biosystems 4500 Q TRAP, www.appliedbiosystems.com.cn/, accessed on 10 February 2021). The analytical conditions were as follows, UPLC: column, Agilent SB-C18 (1.8 µm, 2.1 mm × 100 mm). The mobile phase consisted of solvent A, pure water with 0.1% formic acid, and solvent B, acetonitrile with 0.1% formic acid. Sample measurements were performed with a gradient program that employed the starting conditions of 95% A and 5% B. Within 9 min, a linear gradient to 5% A, 95% B was programmed, and a composition of 5% A, 95% B was kept for 1 min. Subsequently, a composition of 95% A and 5.0% B was adjusted within 1.10 min and kept for 2.9 min. The flow velocity was set as 0.35 mL per minute. The column oven was set to 40 °C. The injection volume was 4 μL. The effluent was alternatively connected to an ESI-triple quadrupole-linear ion trap (QTRAP)-MS.

#### 2.3.3. ESI-Q TRAP-MS/MS

LIT (Linear Ion Trap) and triple quadrupole (QQQ) scans were acquired on a triple quadrupole–linear ion trap mass spectrometer (Q TRAP), AB4500 Q TRAP UPLC/MS/MS system, equipped with an ESI Turbo Ion-Spray interface, operating in positive and negative ion mode and controlled by Analyst 1.6.3 software (AB Sciex). The ESI source operation parameters were as follows: an ion source, turbo spray; source temperature 550 °C; ion spray voltage (IS) 5500 V (positive ion mode)/−4500 V (negative ion mode); ion source gas I (GSI), gas II(GSII), curtain gas (CUR) were set at 50, 60, and 25.0 psi, respectively; the collision-activated dissociation (CAD) was high. Instrument tuning and mass calibration were performed with 10 and 100 μmol/L polypropylene glycol solutions in QQQ and LIT modes, respectively. QQQ scans were acquired as MRM experiments with collision gas (nitrogen) set to medium. DP and CE for individual MRM transitions were done with further DP and CE optimization. A specific set of MRM transitions were monitored for each period according to the metabolites eluted within this period.

#### 2.3.4. Statistical Analysis

R software (R Foundation for Statistical Computing, Vienna, Austria) was used for principal component analysis (PCA) using a “dimensionality reduction” approach to summarize and conclude the presence of metabolic profiles [23]. The metabolite content data were normalized by unit variance scaling, and heat maps were created using the heatmap package in R software to cluster the metabolite accumulation patterns observed between samples [23]. The Pearson correlation coefficient, r, was used as an indicator to assess the biological replicate correlation, and the closer R^2^ was to 1, the stronger the correlation between the two replicate samples. By maximizing the metabolome differences among different variations observed in Orthogonal Partial Least Squares-Discriminant Analysis (OPLS-DA), metabolites with variable importance in projection (VIP) ≥ 1 and FC ≥ 2 or FC ≤ 1/2 in the sample were defined as the metabolites that accumulated between samples differently.

### 2.4. RNA-seq, Annotation, and Data Analysis

Transcriptome sequencing was used to construct 9 libraries of the 3 pericarp samples and 3 repetitive sequences. The transcriptome sequencing method was based on the one used in Xu’s study [19], and in brief, RNA was extracted to determine its purity and concentration and to construct a cDNA library. The Illumina HiSeq platform was then used for sequencing. Low-quality “raw data” were eliminated by removing reads with adapters to obtain “clean data” after sequencing.

Five major databases were used for gene function annotation: National Center for Biotechnology Information (NCBI), Kyoto Encyclopedia of Genes and Genomes (KEGG), nonredundant protein sequences (NR), the SWISS-PROT protein sequence database, and Gene Ontology (GO).

Fragments Per Kilobase of transcript per Million fragments mapped (FPKM) was used to measure the transcription and gene expression levels. DESeq2 (v1.22.2) was used for differential gene expression screening with |log2Fold Change| > 1 and False Discovery Rate (FDR) < 0.05 [24].

### 2.5. qRT-PCR

The expression levels of 12 structural and 3 regulatory genes in the anthocyanin biosynthetic pathway were verified using qRT-PCR. Methods were carried out as described previously [25]. A housekeeping gene (*Actin*) was used as an internal standard. The primers used for qRT-PCR are shown in Appendix A.

## 3. Results

### 3.1. Appearance Characteristic of D. delavayi Fruits, and Quantification of Total Flavonoids and Anthocyanins

In the juvenile stage, the three fruit variations are identical and resemble wild apples in appearance. The fruits gradually show three different skin colors during ripening (Figure 1), with no differences being observed inside the fruit. The flavonoid contents in the three samples ranged from 5.77 mg/g (GP) to 6.35 mg/g (RP) (Figure 2A). The anthocyanin contents in RP, YP, and GP were 36,162.03 ng/g, 31,424.27 ng/g, and 28,315.25 ng/g, respectively (Figure 2B).

### 3.2. Metabolome Data Quality Analysis

The metabolites of the samples showed clear separation between the three different groups, with the two main components in the pericarp, PC1 and PC2, being 46.59% and 34.78%, respectively (Appendix A). The OPLS-DA value was 1 for R^2^Y, demonstrating the reliability of the metabolomics data (Appendix A). Pearson’s correlation factor indicated a remarkable difference among the three sample groups, which is consistent with the PCA data (Appendix A).

### 3.3. The Identification of Differentially Accumulated Metabolites (DAMs) in the Pericarp of Three Different Variations

Through metabolic profiling, we identified 787 compounds (Appendix A) that were able to be classified into 11 classes. The most abundant metabolites were flavonoids followed by phenolic acids, lipids, and other types of metabolites (Figure 3A). Four of the SCMs (2-(formylamino) benzoic acid, ferulic acid, demethyl coniferin, and luteolin-7-O-glucuronide) only exist in RP. The DAM between pairs of samples (GP vs. RP, YP vs. GP, and YP vs. RP) was determined based on VIP ≥ 1 and FC ≥ 2 or FC ≤ 0.5. As expected, a number of DAM differences accumulated between the compared samples, with 239, 283, and 293 DAMs being observed in GP vs. RP, YP vs. GP, and YP vs. RP, respectively (Appendix A). Among the DAMs detected in all of the compared samples, the most enriched KEGG terms were metabolic pathways, the biosynthesis of secondary metabolites, flavonoid biosynthesis, flavone and flavonol biosynthesis, and phenylpropanoid biosynthesis (Figure 3B–D). Comparative analysis of the three groups of DAMs among the green, yellow, and red samples resulted in 68 common metabolites (Figure 3E). There were 44, 48, and 37 metabolites in YP vs. GP, YP vs. RP, and GP vs. RP, respectively, (Figure 3E). Of these, the same differential accumulation pattern (up or down) was maintained among the three variations, and it may contain potential metabolites associated with the pericarp color in *D. delavayi*. These metabolites are from different classes, which suggests that the variation in the color of the peel of *D. delavayi* may be related to other factors such as nutrients and antioxidants. The phenylpropanoid- and flavonoid-related molecules were enriched in these core-conserved metabolites. We hypothesize that differential metabolites in the flavonoid biosynthetic pathway may be the key metabolites that are responsible for the different peel colors observed in the three variations of *D. delavayi* during and after fruit ripening.

### 3.4. Analysis of Anthocyanins in Three Different Variations of D. delavayi

Anthocyanins are important pigments determining the color of plant flowers, fruits, seed coats, etc. There were two anthocyanins that were identified in the pericarp: cyanidin-3-O-galactoside and cyanidin-3-O-glucoside, which accumulated at a rate of 396.071-fold and 445.847-fold in GP vs. RP, 9.188-fold and 7.977-fold in YP vs. GP, and 0.023-fold and 0.018-fold in YP vs. RP, respectively, (Table 1). We speculate that the central metabolite responsible for the color change in *D. delavayi* is anthocyanin.

### 3.5. Transcriptome Sequencing and Annotation

To investigate gene expression in the pericarp among the different variations, the RNA from three samples of wild fruit from *D. delavayi* was used for RNA-seq, with each stage being repeated three times. The transcriptome sequencing of nine samples yielded a total of 66.1 Gb of raw data (Appendix A). The transcript sequences were spliced based on the clean read data, yielding 294,606 transcript assemblies, and a total of 194,405 unigenes were assembled (Appendix A). There were 154,903 (79.68%) unigenes that were annotated in the NR database, and almost half (51.07%) of these unigenes were similar to *Malus domestica* (Figure 4A).

To provide a deeper understanding of the transcriptome differences, experiments were conducted to compare the RNA-seq data from the three different variations of *D. delavayi* fruits, and transcriptome expression was measured using the FPKM value (with FPKM > 0). Among GP vs. RP, YP vs. GP, and YP vs. RP, there were 15,950, 19,820, and 23,079 DEGs, respectively, (Figure 4B–D).

### 3.6. GO Enrichment and KEGG Pathway Analyses of DEGs

To understand the function of DEGs, GO analysis was used to classify them. A total of 47,051 unigenes were annotated to the GO database, with 8097 upregulated and 9482 downregulated unigenes being annotated to the GO library in the sample YP vs. GP. The top-50 enriched GO categories were placed into three clusters, and the most enriched biological process (BP) was cell proliferation. In the cellular component (CC), the genes were mainly enriched in terms of binding. Additionally, in terms of molecular function (MF), genes were mainly enriched in the cell (Appendix A). In the sample YP vs. RP, there were 9849 upregulated unigenes and 8299 downregulated unigenes that were annotated to the GO libraries. In the BP category, cellular processes were the most enriched. In CC, genes were mainly enriched in the cell. Additionally, in terms of MF, genes were mainly enriched in terms of binding (Appendix A). There were 6334 upregulated and 4990 downregulated unigenes in the GP vs. RP sample. The cellular processes of the genes were determined to be the most enriched according to BP classification. In the CC domain, it was determined that enrichment was predominantly in the cell. Binding was clustered in the MF domain (Appendix A).

To classify the DEGs using KEGG pathway enrichment, all the DEGs were enriched in 139 pathways in the three variations of *D. delavayi* (Appendix A). The flavonoid pathway was enriched by KEGG in the three variations (ko00941, 83 genes). In GP vs. RP (Appendix A), the biosynthesis of the secondary metabolites (ko01110, 1505 genes) and ABC transporters (ko02010, 108 genes) as well as phenylpropanoid biosynthesis (ko00940, 171 genes), isoflavonoid biosynthesis (ko00943, 11 genes), anthocyanin biosynthesis (ko00942, 3 genes), the MAPK signaling pathway (ko04016, 287 genes), and flavone and flavonol biosynthesis (ko00944, 5 genes) were enriched. The anthocyanin pathway was enriched (ko00942, 11 genes) in YP vs. GP (Appendix A), and isoflavonoid biosynthesis (ko00943, 20 genes) was enriched in YP vs. RP (Appendix A).

### 3.7. Identification of TFs Related to Anthocyanin Biosynthesis

Among GP vs. RP, YP vs. RP, and YP vs. GP, 809, 1231, and 1140 differential transcriptome factors were identified, respectively. The TFs that were differentially expressed were annotated as AP2/ERF, MYB, bHLH, bZIP, WRKY, NAC, HB-HD-ZIP, MADS, HSF, and other TFs (Appendix A). The research indicates that the biosynthesis of anthocyanins is mainly regulated by MBW proteins and other TFs, which, in turn, directly regulate the upstream and downstream structural genes that are involved in the synthesis process. In GP vs. RP, 79 MYBs and 29 bHLHs were obtained in the pericarp of *D. delavayi*; unfortunately, no WD40 was found. There were 15 MYBs and 2 bHLHs that were expressed in RP. These may be involved in the regulation of anthocyanin biosynthesis in RP. According to previous reports, other TFs (34 NACs, 53 ERFs, 36 WRKYs, 29 bZIPs, and 7 MADS) have also been identified (Appendix A) and may also influence or be involved in the structural and regulatory genes that regulate anthocyanin biosynthesis. In YP vs. RP, 112 MYBs and 55 bHLHs were obtained, and according to FPKM, 15 MYBs and 5 bHLHs were expressed in RP only, in addition to the 64 NACs, 58 ERFs, 42 WRKYs, 51 bZIPs, and 16 MADS that were also found to possibly be involved in anthocyanin regulation (Appendix A).

### 3.8. Analysis of Structural Genes Involved in Anthocyanin Biosynthesis and Gene Expression through qRT-PCR

We constructed a pathway map based on the previously reported anthocyanin biosynthesis pathway, including a heat map of each structural gene expressed in anthocyanins. Further analysis of the data showed that the structural gene expression patterns differed in the different variations of *D. delavayi*. A comparison of the results for the three variations revealed that *C4H*, *4CL*, *CHS*, *DFR*, and *ANS* were highly expressed in GP. However, *F3H*, *F3*′*H*, and *GST* were highly expressed in RP. *F3H* catalyzes the formation of dihydroflavonol from the pre-synthetic substrate naringenin, and dihydroquercetin is catalyzed from *F3′H*. This represents an important precursor and key branching point for the biosynthesis of different types of anthocyanins. We found that one *F3H* and five *F3′H* were all upregulated in the RP. High expression of the structural gene *GST* allows more anthocyanins to be immobilized in the vacuole (Figure 5). To further confirm the RNA-seq results, we performed qRT-PCR analysis on 12 structural genes and 3 regulatory genes in the anthocyanin biosynthetic pathway. The results indicate that the transcriptome data are fully consistent with the expression of pericarp anthocyanin synthesis-related genes (Figure 6).

## 4. Discussion

A phenotype is the most visual manifestation of metabolite accumulation. Peel color is one of the most important characteristics determining a fruit’s commercial value, and it affects the appearance, quality, and storage of fruits [26]. The pericarp of *D. delavayi* is traditionally bright green (GP) or yellow (YP) in appearance, and a red pericarp (RP) has never been reported. Interestingly, we obtained RP samples from a bud variant. This variation can be used as important material for the development and cultivation of new varieties for *D. delavayi*. In recent years, the metabolome and transcriptome have been widely used to breed various plants by combining the gene data in the transcriptome with the phenotypic profiles in the metabolome as well as to enrich metabolic pathways and molecular interactions to screen for key metabolic pathways or genes and to carry out in-depth analyses and validation on metabolites [27,28].

Anthocyanins allow the fruit to manifest bright colors under different pH levels and metal ion conditions [29]. They not only protect plants from environmental stress, but also attract insects and animals that spread the seeds [30]. In addition, anthocyanins have antioxidant, anti-aging, anti-cancer, vision protection, skin beautifying, and other physiological functions [31]. As mentioned in the literature review, color regulation in plant fruit rind is a complex process, and fruit color is an important measure of fruit quality [32]. The peel of *D. delavayi* is rich in bioactive and functional substances, such as alkaloids, flavonoids, organic acids, and phenolic acids, which have antioxidant, vision-protection, aging prevention, immunomodulatory, anti-inflammatory, and anti-cancer effects [33]. Therefore, it is of great importance to study and exploit the bioactive compounds in the rind of *D. delavayi* to breed it and to develop natural health products.

For this study, a total of 787 metabolites were detected from three different variations of *D. delavayi* using metabolomics based on UPLC-MS/MS techniques, and these metabolites were classified into 31 classes during secondary classification. What stands out in the substances found in GP, YP, and RP are anthocyanins, flavonoids, and flavonols, which provide a scientific basis for the study of functional substances and breeding of *D. delavayi*. Anthocyanins play a vital role in fruit coloration and are mainly synthesized through the flavonoid pathway, protecting plants from environmental stresses and providing health benefits to humans [34,35]. Several lines of evidence suggest that the genes *PAL*, *C4H*, *4CL*, *CHI*, and *F3H* play a major role in the early regulation of anthocyanins, while the genes *LODX*/*ANS*, *F3′5′H*, *UFGT*/*3GT*, *F3′H*, and *DFR* play a crucial role in the later stages of anthocyanin synthesis.

RP variation genes (*CHI*, *F3H*, *F3′H*) were highly enriched, while GP variation genes (*C4H*, *4CL*, *CHS*, *DFR*, and *LDOX*/*ANS*) were upregulated. *F3H* functional genes catalyze the formation of dihydroflavonol from substrates as a necessary step in the synthesis of all three branches of anthocyanins and proanthocyanidins (Figure 5). High *F3H* expression was found to promote the accumulation of both anthocyanins and flavonoids [36]. *F3′H* plays a crucial role in catalyzing the formation of dihydroquercetin from dihydroflavonol by binding *DFR* and *UFGT* in *D. delavayi* to produce cyanidin-3-O-galactoside and cyanidin-3-O-glucoside. *F3′H* gene expression determines the synthesis of anthocyanin, which is consistent with tea (*Camellia sinensis*) and *Senecio cruentus* [37,38]. We hypothesized that *F3H* and *F3′H* are key genes in the anthocyanin synthesis pathway of *D. delavayi*. *GST* encodes for a GST protein that transports anthocyanins into the vacuole [39]. *GST* was 418.84-fold higher in RP at a rate that was 418.84-fold higher than in GP, and we hypothesize that the *GST* expression in RP allows more anthocyanins to be transferred to the vacuole to be immobilized, which is important for color differentiation between the variations.

*DFR* can catalyze dihydroflavonol, dihydromyricetin, and dihydroquercetin to form leucopelargonidin, leucodelphinidin, and leucocyanidin, which are the precursors of *ANS* (Figure 5). *ANS* and *DFR* work together to catalyze the synthesis of cyanidin, and it is theorized that relatively high *ANS* and *DFR* activity probably provide more cyanidin for anthocyanin synthesis [40,41]. However, the present study found that the expression of *DFR* in GP was higher than that of YP and higher than that of RP. It is hypothesized that this inconsistency in gene FPKM and anthocyanin content may be due to selective catalytic action by *DFR*. Some evidence suggests that substrate-specific *DFR* was found to be ineffective in reducing dihydrokaempferol in studies on peony and grape hyacinth (*Muscari* spp.) [42,43]. Another important finding is the altered substrate specificity of *DFR* observed in buckwheat studies [44]. We speculate that it is the selective catalysis of the *DFR* gene that allows the increased conversion of dihydroflavonol to cyanidin.

Previous studies have shown that cyanidin-3-O-galactoside is the main coloring substance in strawberries [45], red pears [46], and apples [13,47]. This may be because plants in the same family have similar regulatory genes and reproductive organs (flowers) for flowering and fruiting. Furthermore, in studies on blood orange [48], *Litchi chinensis* [49], and longan [25], it was found that the cyanidin 3-glucoside content increased as the peel became redder in color, eventually becoming the main coloring substance. These studies suggest some similarities in the coloring regulation of anthocyanins. Cyanidin-3-O-galactoside and cyanidin-3-O-glucoside were detected in all the samples, with the highest levels of both anthocyanins being observed in RP. Intermediate levels were found in YP, and the lowest levels were found in in GP. The cyanidin-3-O-galactoside and cyanidin-3-O-glucoside contents in RP fruit were 396.1-fold and 445.8-fold higher than they were in in GP; furthermore, the content of both anthocyanins was up-regulated by 9.2- and 8.0-fold, respectively, in RP compared to GP. Therefore, we hypothesize that the bright red color of RP fruit is due to the abundance of cyanidin-3-O-galactoside and cyanidin-3-O-glucoside in its pericarp.

The biosynthesis of anthocyanins in plants is mainly regulated by the MBW complex and other TFs [50]. Based on the FPKM of the genes, we identified 15 MYBs and 2 bHLHs that were only expressed in RP among GP vs. RP and 15 MYBs and 5 bHLHs that were only expressed in RP among YP vs. RP. In red pear, the TFs PyMYB10, PyMYB114, and PybHLH3 were identified to regulate the expression of the *PyDFR*, *PyANS*, and *PyUFGT* genes [20]. However, in this study, with the activation of MBW regulation, the FPKM of the genes *DFR* and *ANS* decreased, but more anthocyanins accumulated, a phenomenon similar to the one observed in longan [25]. We also identified the specific expression levels of WRKY, NAC, bZIP, ERF, and MADS in RP, as they have been shown to regulate anthocyanin biosynthesis either directly or in an indirect manner by binding to the MYB promoter in model plants and in other fruits or through protein–protein interactions. Such variables need to be taken into account and studied further to reveal the regulatory relationship between this phenomenon and genes. All these findings will help to clarify anthocyanin biosynthesis in *D. delavayi* in terms of molecular mechanisms and regulatory networks and will provide a biological basis for the breeding of new varieties.

## 5. Conclusions

In our study, metabolomics based on UPLC-MS/MS technology and transcriptomic approaches based on the Illumina Novaseq platform were used to investigate the mechanism of pericarp coloration at maturity in different variations of *D. delavayi*. Flavonoids were the major secondary metabolites found in the pericarp of *D. delavayi*, and cyanidin-3-O-galactoside and cyanidin-3-O-glucoside were the major secondary metabolites found for pericarp coloration. The enzyme activities related to anthocyanin synthesis showed different patterns of variation in the different variations of *D. delavayi*. The present study has largely improved our understanding of flavonoid-related metabolite components in *D. delavayi* and has provided important reference values for breeders to improve the pericarp coloration of *D. delavayi*.

## Figures and Tables

**Figure 1 foods-11-02899-f001:**
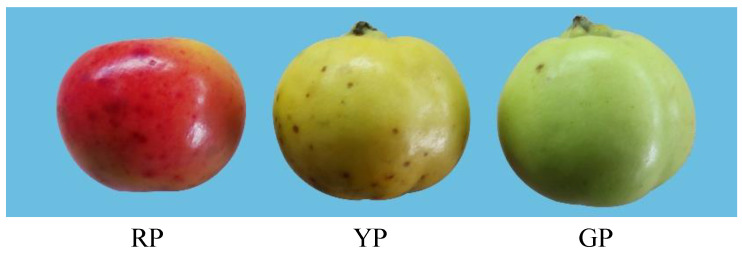
The different variations of *D. delavayi*: RP = red peel *D. delavayi*; YP = yellow peel *D. delavayi*; GP = green peel *D. delavayi*.

**Figure 2 foods-11-02899-f002:**
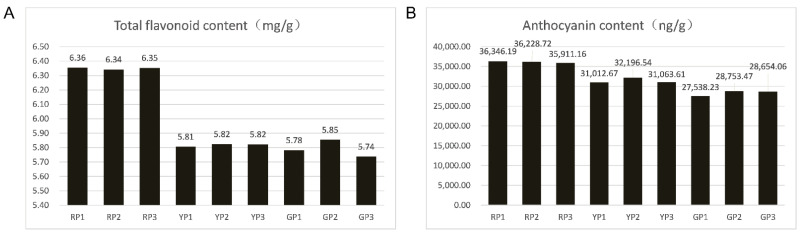
Quantification of bioactive substances: (**A**) total flavonoid content; (**B**) anthocyanin content.

**Figure 3 foods-11-02899-f003:**
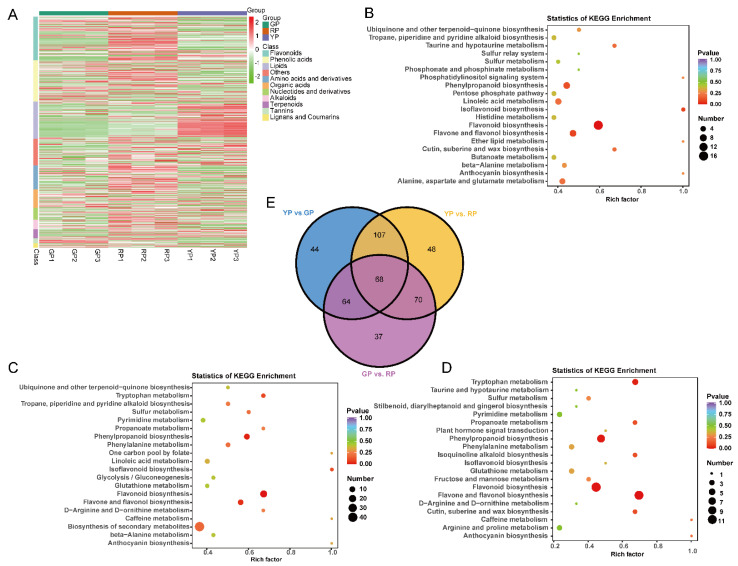
DAM identification: (**A**) heat map of DAM classification and KEGG enrichment scatter plots: (**B**) YP vs. GP; (**C**) YP vs. RP; (**D**) GP vs. RP; (**E**) Venn of DAMs among the three variations.

**Figure 4 foods-11-02899-f004:**
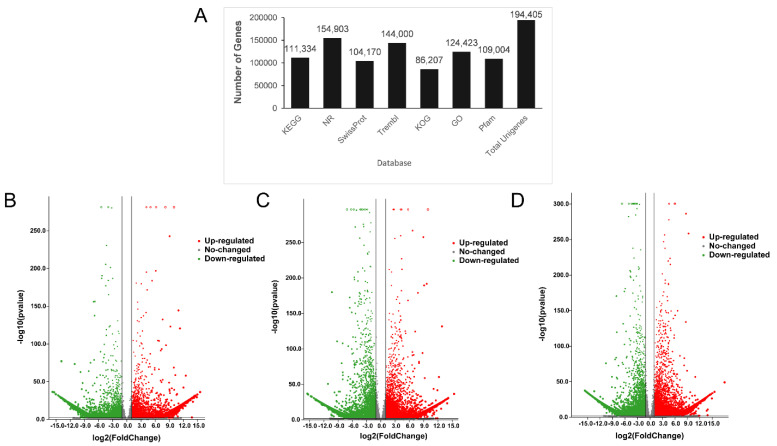
Identification of differential genes: (**A**) annotation by the seven databases; volcano plots of differential gene expression analysis: (**B**) GP vs. RP, (**C**) YP vs. GP, (**D**) YP vs. RP.

**Figure 5 foods-11-02899-f005:**
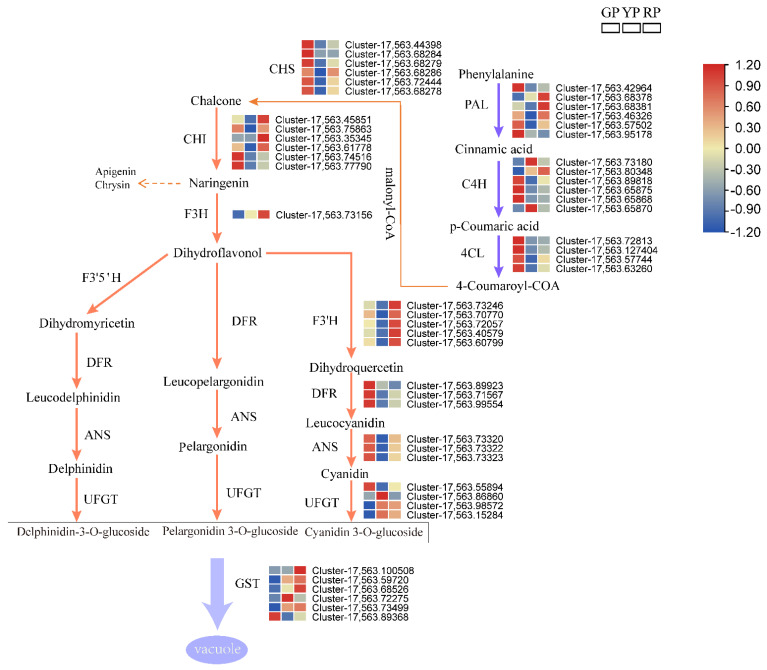
Expression levels of structural genes involved in the anthocyanin biosynthesis pathway in the fruit of the *D. delavayi*. The heatmap represents the expression of corresponding structural genes in three *D. delavayi* variations, and in the heatmap, the gradient from blue to red indicates the expression levels of structural genes ranging from low to high. Enzymes in this pathway are annotated as follows: *PAL*, phenylalanine ammonia lyase; *C4H*, cinnamate 4-hydroxylase; *4CL*, 4-coumarate: CoA ligase; *CHS*, chalcone synthase; *CHI*, chalcone isomerase; *F3H*, flavanone 3-hydroxylase; *F3′H*, flavonoid 3′-hydroxylase; *F3′5′H*, flavonoid 3′,5′-hydroxylase; *DFR*, dihydroflavonol 4-reductase; *ANS*, anthocyanidin synthase; *UFGT*, UDP-glucose: flavonoid 3-O-glucosyltransferase; *3GT*, UDP-glucose: anthocyanidin 3-glucosyltransferase; *GST*, glutathione S-transferase.

**Figure 6 foods-11-02899-f006:**
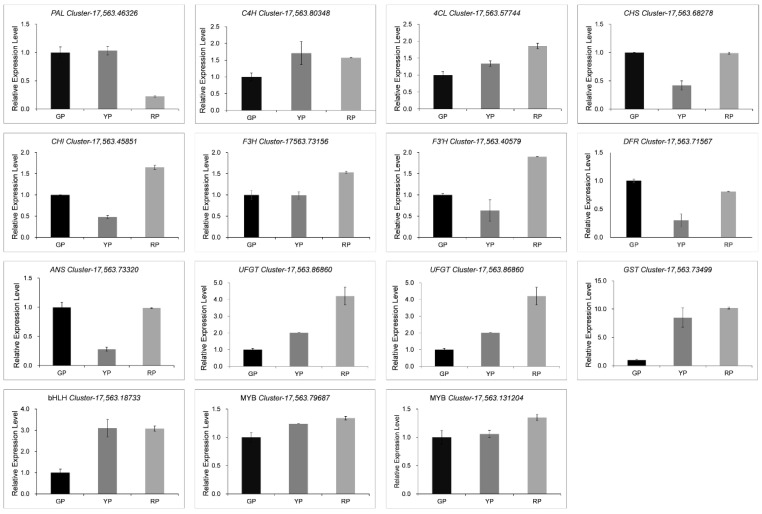
Analysis of the expression levels of key structural genes and transcription factors during anthocyanin biosynthesis in *D. delavayi*.

**Table 1 foods-11-02899-t001:** Differential accumulation of anthocyanins in the pericarp of the three variations.

Compounds	Ion Abundance			Fold Change		
	GP	YP	RP	GP vs. RP	YP vs. GP	YP vs. RP
Cyanidin-3-O-galactoside	4.55 × 10^5^	1.96 × 10^7^	1.80 × 10^8^	396.071	0.023	9.188
Cyanidin-3-O-glucoside	3.89 × 10^5^	2.17 × 10^7^	1.74 × 10^8^	445.847	0.018	7.977

## Data Availability

The data that support the findings of this study are openly available at NCBI at https://www.ncbi.nlm.nih.gov/sra/PRJNA751484, reference number PRJNA751484, and all data generated or analyzed during this study are included in this published article (and its Appendix A).

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
