# Peer review of "Transcriptomic and Metabolomic Analyses Providing Insights into the Coloring Mechanism of *Docynia delavayi"

_foods, 2022, doi:10.3390/foods11182899_

Round 1
Reviewer 1 Report
1. Authors don't consider genetic variation for fruit peel coloration, including bud mutation. There are a lot of the related studies in genetic research fields. But, This study did not reproduce previous studies or discover new things. Thus, this study have to be reorganized with new descriptions.
2. The manuscript is not readable since there are numerous grammar errors (singular vs plural, uppercase vs lowercase, phrase vs sentence, and spacing). I strongly recommend that when submitting the manuscript in journals, the authors proofread the manuscript with colleagues who are accustomed to English writing or professional English services.
Author Response
Dear reviewer,
Thank you very much for your comments and professional advice. These opinions help to improve the academic rigor of our article. Based on your suggestion and request, we have made corrected modifications to the revised manuscript. Meanwhile, the manuscript had to be reviewed and edited by the language services of MDPI. We hope that our work can be improved again. Furthermore, we would like to show the details as follows:
Point 1: Authors don't consider genetic variation for fruit peel coloration, including bud mutation. There are a lot of the related studies in genetic research fields. But, This study did not reproduce previous studies or discover new things. Thus, this study have to be reorganized with new descriptions.
Response 1: Thanks for your kind and valuable suggestion.
According to your advice, we have indicated the genetic traits of the pericarp in the section of the Introduction.
Lines 41-51,” The pericarp color leaves a visual impression on consumers and is an important indicator of fruit selection. The bright color of the pericarp attracts animals to spread its seeds, and the pericarp also helps the plant to resist abiotic stress [3, 4]. In prolonged investigations and research, it was found that D. delavayi has rich phenotypic variation, which is especially evident in the color of its fruit peels. D. delavayi traditionally has a bright green pericarp (GP) or yellow pericarp (YP) (Figure 1). Over the course of our investigations, we found a red bud variant and obtained stable genetic traits (fruit ripening to red pericarp) through asexual reproduction (Figure 1). Understanding the genetic basis of the traits associated with the color of the skin of D. delavayi fruit will help us to breed it to be more attractive and healthier for consumers. ”
According to your advice, we have redescribed the discussion section.
Lines 391-406, “Previous studies have shown that cyanidin-3-O-galactoside is the main coloring substance in strawberries [45], red pears [46], and apples [13, 47]. This may be because plants in the same family have similar regulatory genes and reproductive organs (flowers) for flowering and fruiting. Furthermore, in studies on blood orange [48], Litchi chinensis [49], and longan[25], it was found that the cyanidin 3-glucoside content increased as the peel became redder in color, eventually becoming the main coloring substance. These studies suggest some similarities in the coloring regulation of anthocyanins. Cyanidin-3-O-galactoside and cyanidin-3-O-glucoside were detected in all of the samples, with the highest levels of both anthocyanins being observed in RP. Intermediate levels were found in YP, and the lowest levels were found in in GP. The cyanidin-3-O-galactoside and cyanidin-3-O-glucoside contents in RP fruit were 396.1-fold and 445.8-fold higher than they were in in GP; furthermore, the content of both anthocyanins was up-regulated by 9.2- and 8.0-fold, respectively, in RP compared to GP. Therefore, we hypothesize that the bright red color of RP fruit is due to the abundance of cyanidin-3-O-galactoside and cyanidin-3-O-glucoside in its pericarp.”
Point 2: The manuscript is not readable since there are numerous grammar errors (singular vs plural, uppercase vs lowercase, phrase vs sentence, and spacing). I strongly recommend that when submitting the manuscript in journals, the authors proofread the manuscript with colleagues who are accustomed to English writing or professional English services.
Response 2: We apologize for the language problem, and we have submitted the manuscript to the MDPI editing services for further improvement and review by native English speakers.
Thanks for all your comments and suggestions.

Reviewer 2 Report
The article ‘The transcriptomic and metabolomic analysis provides insights into the Coloring Mechanism of Docynia delavayi‘ presented for review is interesting. However, some issues need to be clarified or supplemented. The comments are included below:
Title
- The title is worded correctly and accurately reflects the content.
Abstract
- The abstract is clear and adequate.
1. Introduction
- I suggest developing the introduction. It is currently too short.
2. Materials and methods
2.2. Quantification of total flavonoids and anthocyanins
- How the samples were prepared for analysis? How the anthocyanins were extracted?
3. Results
- How was the qualitative analysis and identification of the anthocyanins presented in Table 1 carried out. Please describe it in the methodology.
- Whether the method used allowed to determine the content of individual anthocyanin fractions?
- Figure 5 is difficult to read.
- Figure 7. The title of the chart should start with a capital letter.
4. Discussion
- The discussion is appropriate.
5. Conclusions
- In the opinion of the reviewer, the conclusions are too general. This section needs to be completed.
Another
- I suggest you correct the text for punctuation errors.
Author Response
Dear reviewer,
Thank you very much for your comments and professional advice. These opinions help to improve the academic rigor of our article. Based on your suggestion and request, we have made corrected modifications to the revised manuscript. Meanwhile, the manuscript had to be reviewed and edited by the language services of MDPI. We hope that our work can be improved again. Furthermore, we would like to show the details as follows:
Point 1: Title
- The title is worded correctly and accurately reflects the content.
Point 2: Abstract
- The abstract is clear and adequate.
Response1/2: Thanks for your positive comments.
Point 3: 1. Introduction
- I suggest developing the introduction. It is currently too short.
Response 3: According to your suggestion, we have developed the introduction section with the following details:
Lines 31-34, “D. delavayi is highly adaptable, and its fruits are rich in nutrients, making it a famous local fruit. The destruction of the environment, the loss of wild resources, and the failure to reasonably apply germplasm resources has affected the development D. delavayi cultivation.”
Lines 40-51, “The pericarp color leaves a visual impression on consumers and is an important indicator of fruit selection. The bright color of the pericarp attracts animals to spread its seeds, and the pericarp also helps the plant to resist abiotic stress [1, 2]. In prolonged investigations and research, it was found that D. delavayi has rich phenotypic variation, which is especially evident in the color of its fruit peels. D. delavayi traditionally has a bright green pericarp (GP) or yellow pericarp (YP) (Figure 1). Over the course of our investigations, we found a red bud variant and obtained stable genetic traits (fruit ripening to red pericarp) through asexual reproduction (Figure 1). Understanding the genetic basis of the traits associated with the color of the skin of D. delavayi fruit will help us to breed it to be more attractive and healthier for consumers. ”
- Tewksbury, J. J.; Reagan, K. M.; Machnicki, N. J.; Carlo, T. A.; Haak, D. C.; Peñaloza, A. L.; Levey, D. J., Evolutionary ecology of pungency in wild chilies. Proceedings of the National Academy of Sciences of the United States of America 2008, 105, (33), 11808-11811, doi; 10.1073/pnas.0802691105.
- Zamljen, T.; Jakopič, J.; Hudina, M.; Veberič, R.; Slatnar, A., Influence of intra and inter species variation in chilies (Capsicum spp.) on metabolite composition of three fruit segments. Sci Rep 2021, 11, (1), 4932, doi; 10.1038/s41598-021-84458-5.
Point 4: 2. Materials and methods
2.2. Quantification of total flavonoids and anthocyanins
- How the samples were prepared for analysis? How the anthocyanins were extracted?
Response 4: According to your advice, details of the experimental samples preparation are as follows:
Lines 80-93, The modified version is “In our study, we collected three types (RP, YP, and GP) of mature D. delavayi (Figure 1). Samples were selected from plants with normal growth and no obvious pests or diseases. We selected 30 fruits of equal maturity and with a similar color, shape, and size. The fruits were rinsed with sterile water and dried. Fruit peel samples (0.1 cm thick) were collected from several D. delavay (Rosaceae) trees that were approximately 30-years-old and growing in Lancang ((22°29′55″N; 100°14′59″E, 1950m) Yunnan, China) in October 2020. The fruit peels from each of the three fruit trees were evenly mixed as one biological replicate to create a total of three biological replicates. Three biological replicates were selected for each plant type. Fruit peels were cut off and immediately frozen in liquid nitrogen and stored at -80°C until further analyses were carried out.”
According to your advice, The anthocyanin content was assayed according to Li et al. [3]. The corresponding descriptions in lines 102-110.
- Li, F.; Wu, B.; Yan, L.; Qin, X.; Lai, J., Metabolome and transcriptome profiling of Theobroma cacao provides insights into the molecular basis of pod color variation. J. Plant Res. 2021, 134, (6), 1323-1334, doi; 10.1007/s10265-021-01338-9.
Point 5: 3. Results
- How was the qualitative analysis and identification of the anthocyanins presented in Table 1 carried out. Please describe it in the methodology.
Response 5:
The anthocyanins in Table 1 were integrated from metabolomic data, The corresponding descriptions in lines 112-119.
Point 6:- Whether the method used allowed to determine the content of individual anthocyanin fractions?
Response 6: This is allowed, and this analysis method is reported through the published literature to determine anthocyanin content and individual anthocyanin fractions [4-8].
- Dong, T.; Han, R.; Yu, J.; Zhu, M.; Zhang, Y.; Gong, Y.; Li, Z., Anthocyanins accumulation and molecular analysis of correlated genes by metabolome and transcriptome in green and purple asparaguses (Asparagus officinalis, L.). Food Chem 2019, 271, 18-28, doi; 10.1016/j.foodchem.2018.07.120.
- Liu, Y.; Lv, J.; Liu, Z.; Wang, J.; Yang, B.; Chen, W.; Ou, L.; Dai, X.; Zhang, Z.; Zou, X., Integrative analysis of metabolome and transcriptome reveals the mechanism of color formation in pepper fruit (Capsicum annuum L.). Food Chem 2020, 306, 125629, doi; 10.1016/j.foodchem.2019.125629.
- Shi, Q.; Du, J.; Zhu, D.; Li, X.; Li, X., Metabolomic and transcriptomic analyses of anthocyanin biosynthesis mechanisms in the color mutant Ziziphus jujuba cv. Tailihong. J Agr Food Chem 2020, 68, (51), 15186-15198, doi; 10.1021/acs.jafc.0c05334.
- Xu, J.; Yan, J.; Li, W.; Wang, Q.; Wang, C.; Guo, J.; Geng, D.; Guan, Q.; Ma, F., Integrative analyses of widely targeted metabolic profiling and transcriptome data reveals molecular insight into metabolomic variations during apple (Malus domestica) fruit development and ripening. Int J Mol Sci 2020, 21, (13), 4797, doi; 10.3390/ijms21134797.
- Zhang, Z.; Tian, C.; Zhang, Y.; Li, C.; Li, X.; Yu, Q.; Wang, S.; Wang, X.; Chen, X.; Feng, S., Transcriptomic and metabolomic analysis provides insights into anthocyanin and procyanidin accumulation in pear. BMC Plant Biol 2020, 20, (1), 129, doi; 10.1186/s12870-020-02344-0.
Point 7:- Figure 5 is difficult to read.
Response 7: We redrew the image and added it to the revised supplementary information file.
Point 8:- Figure 7. The title of the chart should start with a capital letter.
Response 8: Revised accordingly.
Point 9: 4. Discussion
- The discussion is appropriate.
Response 9: Thanks for your positive comments.
Point 10: 5. Conclusions
- In the opinion of the reviewer, the conclusions are too general. This section needs to be completed.
Response 10: According to your suggestion, we further summarized the genetic variation of the color of the Docynia delavayi as follows:
Lines 444-454, “In our study, metabolomics based on UPLC-MS/MS technology and transcriptomic approaches based on the Illumina Novaseq platform were used to investigate the mechanism of pericarp coloration at maturity in different variations of D. delavayi. Flavonoids were the major secondary metabolites found in the pericarp of D. delavayi, and cyanidin-3-O-galactoside and cyanidin-3-O-glucoside were the major secondary metabolites found for pericarp coloration. The enzyme activities related to anthocyanin synthesis showed different patterns of variation in the different variations of D. delavayi. The present study has largely improved our understanding of flavonoid-related metabolite components in D. delavayi and has provided important reference values for breeders to improve the pericarp coloration of D. delavayi.”
Point 11: Another
- I suggest you correct the text for punctuation errors.
Response 11: Sorry for the language problems. We checked and modified all punctuation in the manuscript. We have now worked on both language and readability and have also involved native English speakers in language corrections.
Thanks for all your comments and suggestions.
